# Earlier Bedtime and Effective Coping Skills Predict a Return to Low-Risk of Depression in Young Adults during the COVID-19 Pandemic

**DOI:** 10.3390/ijerph191610300

**Published:** 2022-08-18

**Authors:** Qingyu Zhao, Kevin Wang, Orsolya Kiss, Dilara Yuksel, Massimiliano de Zambotti, Duncan B. Clark, David B. Goldston, Kate B. Nooner, Sandra A. Brown, Susan F. Tapert, Wesley K. Thompson, Bonnie J. Nagel, Adolf Pfefferbaum, Edith V. Sullivan, Kilian M. Pohl, Fiona C. Baker

**Affiliations:** 1Department of Psychiatry and Behavioral Sciences, Stanford University, Stanford, CA 94304, USA; 2Department of Computer Science, Stanford University, Stanford, CA 94305, USA; 3Center for Health Sciences, SRI International, Menlo Park, CA 94025, USA; 4Department of Psychiatry, University of Pittsburgh, Pittsburgh, PA 15260, USA; 5Department of Psychiatry and Behavioral Sciences, Duke University School of Medicine, Durham, NC 27708, USA; 6Department of Psychology, University of North Carolina Wilmington, Wilmington, NC 28403, USA; 7Department of Psychiatry, University of California, San Diego, CA 92093, USA; 8Population Neuroscience and Genetics Lab, University of California, San Diego, CA 92093, USA; 9Department of Radiology, University of California, San Diego, CA 92093, USA; 10Departments of Psychiatry and Behavioral Neuroscience, Oregon Health & Science University, Portland, OR 97239, USA

**Keywords:** COVID-19, young adults, depressive symptoms, sleep, coping

## Abstract

To determine the persistent effects of the pandemic on mental health in young adults, we categorized depressive symptom trajectories and sought factors that promoted a reduction in depressive symptoms in high-risk individuals. Specifically, longitudinal analysis investigated changes in the risk for depression before and during the pandemic until December 2021 in 399 young adults (57% female; age range: 22.8 ± 2.6 years) in the United States (U.S.) participating in the National Consortium on Alcohol and NeuroDevelopment in Adolescence (NCANDA) study. The Center for Epidemiologic Studies Depression Scale (CES-D-10) was administered multiple times before and during the pandemic. A score ≥10 identified individuals at high-risk for depression. Self-reported sleep behavior, substance use, and coping skills at the start of the pandemic were assessed as predictors for returning to low-risk levels while controlling for demographic factors. The analysis identified four trajectory groups regarding depression risk, with 38% being at low-risk pre-pandemic through 2021, 14% showing persistent high-risk pre-pandemic through 2021, and the remainder converting to high-risk either in June 2020 (30%) or later (18%). Of those who became high-risk in June 2020, 51% were no longer at high-risk in 2021. Logistic regression revealed that earlier bedtime and, for the older participants (mid to late twenties), better coping skills were associated with this declining risk. Results indicate divergence in trajectories of depressive symptoms, with a considerable number of young adults developing persistent depressive symptoms. Healthy sleep behavior and specific coping skills have the potential to promote remittance from depressive symptoms in the context of the pandemic.

## 1. Introduction

Young adulthood encompasses the transition from late adolescence to adulthood and involves profound behavioral, psychosocial, and neurostructural changes [1]. Factors influencing neurofunctional maturation of young adults include brain developmental processes, ongoing identity formation [2], and the achievement of traditional adult roles in romantic and professional life [3]. This period of life is also associated with elevated risk-taking behaviors [4] and increased risk of substance misuse [3] and depressive symptoms [5].

Notably, socially unpredictable environments and stress induced by the COVID-19 pandemic have resulted in decreased physical and mental health of young adults and adolescents. They faced limited opportunities for in-person interaction, financial strain [6,7,8,9], and constricted access to external support systems [10]. Among many other studies documenting an increase in depressive symptoms among young adults during the COVID-19 pandemic [11,12], our prior analysis of the participants from the National Consortium on Alcohol and Neurodevelopment in Adolescence (NCANDA) study indicated that risk for depression in young adults tripled in June 2020 compared to risk for depression in eight pre-pandemic years [13].

Beyond pre-pandemic risk factors for depression, such as internalizing problems [14,15,16], some behaviors during the pandemic have contributed to greater risk for depression. For example, substance use has increased in some young adults, especially in women [17], contributing to mental health issues [18]. Poor sleep quality [19] and disrupted sleep, longer sleep latency, and higher daytime dysfunction [20] were prevalent among young adults during the COVID-19 outbreak, contributing to higher levels of depression than those reported in older adults. Sleep difficulties are well-documented risk factors for depressed mood, implicating sleep as a promising modifiable factor for preventing mood deficits [21] and improving depression outcomes [22]. Another crucial factor in the prevention and recovery of psychopathology in young people that could be critical in the context of the pandemic is adaptive coping strategies in response to life stress [23,24,25]. Longitudinal analyses of 1251 adolescents [24] revealed that the association between poorer coping strategies and depressive symptoms increased after the age of 17.5 years. In a sample of 881 first-year college students in the Netherlands, maladaptive coping (such as denial, venting, substance use, and self-blame) mediated the trajectory of depression and anxiety at the beginning of the pandemic [26]. Compared to older adults, the different coping strategies of younger individuals had negative effects, rendering them vulnerable to developing psychopathology and stressful events [27].

Although much effort has been devoted to studying the effects of the onset of COVID-19 on psychological wellbeing, few studies have measured changes in youth depression symptoms and potential recovery across the pandemic period. Kujawa et al. [28] reported high rates of depression in young adults in the US in May 2020 (45.1% of the sample was above the clinical cutoff for depression), decreasing one month later (35.9% meeting the cutoff). A study [29] of Chinese college students conducted between February and June 2020 identified five sub-cohorts with distinct depression trajectories: resistance, recovery, delayed-dysfunction, chronic-dysfunction, and relapsing/remitting. Their assessments showed that coping skills, higher social support, and better family functioning were related to improvements in depressive symptoms during those early months of the pandemic.

While our prior study [13] aimed to identify predisposing factors associated with a higher risk for persistent distress or mental health decline during the onset of COVID-19, the aim of the current analysis was to reveal factors that aid in returning to low-levels of risk among young adults with pandemic-related high-risk for depression. To this end, we expanded our analysis of the NCANDA dataset (age range: 18 to 28.6 years at the start of the pandemic) from comparing risk for depression in 2020 with pre-pandemic levels (median visit June 2019) [13] to examining heterogeneity in longer-term trajectories of depression symptoms through the end of 2021. Our second aim was to identify factors that differentiated participants who were at high-risk for depression at the start of the pandemic and who either later remitted or stayed at persistently high risk for depression. Recognition of specific factors that predict remittance from depressive symptoms can encourage embracing adaptive processes in response to COVID-19 or other social and environmental challenges.

## 2. Methods

### 2.1. Participants and Procedures

The NCANDA cohort comprises 831 participants (ages 12 to 21 years at baseline) recruited between 2013 and 2014 across five U.S. collection sites. School- and community-based recruitment at each site was designed to reflect the local racial/ethnic distribution of their area with equal sex proportions across the age range (see [30] for details); 83% of the participants had limited or no history of alcohol use at baseline, and the remaining participants exceeded drinking thresholds for their age. Participants engaged in annual follow-up visits that were distributed across each calendar year. Informed consent was provided by parents/legal guardians for minor participants, who also gave written assent. The Institutional Review Boards of each site approved the study. For each participant, the most recent visit before 1 March 2020 was considered their pre-COVID-19 assessment. This cutoff date was chosen because shelter-in-place orders began to be issued across the U.S. thereafter.

Four online surveys were distributed to participants in 2020 and 2021 to determine the effects of the COVID-19 pandemic: Survey 1, June 2020; Survey 2, December 2020; Survey 3, June 2021; and Survey 4, December 2021. Data were acquired from 399 emerging adults (227 women ages 18–28.6 years at Survey 1) who completed all four COVID-19 surveys regardless of drinking status. The pre-COVID-19 visits of these 399 adults ranged from July 2013 to February 2020, with the median date in June 2019 (Appendix A). Compared to the remaining 432 NCANDA participants who missed at least one of the surveys, participants included in this analysis were more likely to be female (Chi-square test, *p* = 0.001) and less likely to be African American (Chi-squared test, *p* = 0.03). They had higher family socioeconomic status (*t*-test, *p* = 0.01) than those not included (n = 432). We used parental years of education as an indicator of family socioeconomic status because income varies widely with geography [31,32]. Table 1 summarizes the demographic characteristics of the sample.

### 2.2. Measures

Measures of depressive symptoms: To examine the effect of the pandemic on the trajectory of risk of depression, we used the Center for Epidemiologic Studies Depression Scale (CES-D-10) score of each participant measured at the pre-COVID-19 assessment and each of the surveys completed during the pandemic. The CES-D-10 is a shortened, validated version of the 20-item CES-D [33]. It assesses depressive symptoms in the prior 7 days using 10 items, with responses ranging from “rarely or none of the time” (score of 0) to “all of the time” (score of 3). Total scores range from 0 to 30, with higher scores indicating the presence of more depressive symptoms. The CES-D-10 has good psychometric properties [34,35] and is sensitive to change [36]. Although the CES-D-10 is not intended to diagnose major depressive disorder (MDD), when it is used as a screening tool, a cutoff score ≥10 has been suggested as indicating significant depressive symptoms [37], which we refer to as high-risk for depression.

Measures of alcohol use, sleep, and coping strategies: At each COVID-19 survey, self-reported alcohol use was assessed with questions from the Customary Drinking and Drug-use Record (CDDR) questionnaire [38] including frequency in the past 30 days and quantity in an average 24-h period. Participants also completed the subset of items from the PSQI (Pittsburgh Sleep Quality Index) [39] that asked about bedtimes, wake-up times, sleep duration, and sleep quality in the past month. Sleep quality was assessed with the question, “During the past month, how would you rate your sleep quality overall?”, with options ranging from very good to very bad. The choice of using this single rating to encode sleep quality is due to its strong correlation with the total PSQI score in the NCANDA sample (Hasler et al., 2017). Coping was quantified using the Brief Resilient Coping Scale (BRCS) [40], which measures the tendency to cope with stress adaptively. The BRCS consists of 4 items, including self-descriptive statements (e.g., “I believe I can grow in positive ways by dealing with difficult situations”) that participants rate on a five-point Likert-type scale. The total score ranges from 4 to 20, with higher scores reflecting more resilient coping.

### 2.3. Statistical Analysis

Significant changes in CES-D-10 scores between any two consecutive COVID-19 surveys and from the pre-COVID-19 survey to each COVID-19 survey were examined by two-tailed paired t-tests. Bonferroni correction for the 7 comparisons required *p* < 0.007.

For the 119 participants who were below the risk threshold for clinical depression at their last pre-COVID-19 visit but exceeded the threshold (CES-D-10 ≥ 10) [37] at COVID-19 Survey 1, a logistic regression model predicted which of them would return to low-risk in 2021, i.e., the average CES-D-10 score across the two 2021 surveys was below 10.

We first applied a base logistic regression model to differentiate the two groups using only nuisance predictors at the start of the pandemic (i.e., Survey 1), which were age, sex, age-by-sex interaction, site, race, ethnicity, family socioeconomic status, and education status (whether a participant was enrolled in any type of school, educational classes, or coursework). We then applied a complete model by adding to the base model the CES-D-10 score in June 2020 and the interactions with age and sex of the following 6 predictors measured in June 2020: coping scale score, frequency and quantity of alcohol consumption, bedtime, sleep duration, and sleep quality. For both models, non-categorical variables were transformed to z-scores. Because of their skewed distributions, the alcohol consumption variables were log-transformed before being entered into the regression. The goodness of fit for the complete model was compared to that of the base model using the Likelihood Ratio Test (*p* < 0.05 as significance level), which automatically corrected for the number of additional predictors in the complete model. A significantly higher goodness of fit (*p* < 0.05) would suggest those additional predictors contributed to the prediction of returning to low-risk status.

To identify the subset of individual factors contributing to the prediction, we selected those predictors with *p* < 0.05 in the complete logistic regression. If the age or sex interaction with a modifiable factor was a significant predictor, a simple t-test examined the difference in that factor between the returning to low-risk and persistent high-risk participants within each sex and age group (divided by the median age of all participants).

Lastly, the complete model was retested in two settings. To ensure the prediction was not confounded by the initial CES-D-10 score in June 2020, we first retested the complete model on 49 young adults returning to low-risk and 49 persistent high-risk individuals. These two sub-groups were selected from the 119 participants by matching their CES-D-10 scores in June 2020 (*p* = 0.171) (see Supplement for matching procedure). In the second setting, we used the more conservative threshold of CES-D-10 ≤ 7 for returning to low-risk participants and CES-D-10 ≥ 10 for persistent high-risk participants in 2021.

## 3. Results

Figure 1a shows the distribution of the CES-D-10 score for the 399 NCANDA participants measured at their most recent pre-COVID-19 visit and the four COVID-19 surveys. Consistent with our previous report [13], the CES-D-10 score increased significantly in June 2020 compared to pre-COVID-19 visits (*p* < 0.001) and then again between June and December of 2020 (*p* = 0.003). The CES-D-10 score then significantly declined in June 2021 (Surveys 2 vs. 3: *p* < 0.001) and then stabilized in December 2021 (Surveys 3 and 4: *p* = 0.399). In neither June nor December 2021 did the CES-D-10 score return to the pre-COVID-19 level (*p* < 0.001).

Based on the risk threshold for clinical depression (CES-D-10 ≥ 10), trajectories of CES-D-10 from pre-COVID-19 to COVID-19 surveys differed substantially across participants (Figure 1b). Over 38% of participants remained below the low-risk level for depression across all assessments (n = 153), 29.8% of the participants converted to high-risk in June 2020 (n = 119), 18.3% converted either in December 2020 or in 2021 (n = 73), and the remaining participants were at high-risk for depression before the pandemic and remained at high-risk throughout the pandemic (Figure 1b).

Among the 119 participants who exceeded the depression threshold in June 2020 (yellow in Figure 1b), 61 returned to below the high-risk level in 2021 and 58 remained at the high-risk level (Figure 1c). None of the nuisance variables in the base logic regression model were significant predictors of this group assignment (Table 2).

The complete model resulted in significantly higher goodness of fit (*p* = 0.022) than the base model (Table 3). The returning to low-risk group included a higher percentage of Hispanic participants (n = 9) than the persistent group (n = 4). Compared to the persistent participants, the participants returning to low-risk had earlier bedtimes in June 2020 (38 min on average, *p* = 0.018, Table 3) despite the difference between the two groups being insignificant (6 min on average, *p* = 0.591) before the COVID-19 pandemic. Another significant predictor was the age-by-coping interaction (*p* = 0.010, Table 3). The post-hoc t-tests within the older and younger cohorts (which were partitioned according to their median age at 22.2 years) also supported that the coping scale score of the group with declining CES-D scores was significantly higher than the persistent group for older (*p* = 0.047) but not for younger participants (*p* = 0.125) (Figure 2b).

Lastly, reapplying the complete model to the two sub-groups of 49 participants matched with respect to CES-D-10 score in June 2020 revealed that earlier bedtime and better coping strategy in older participants endured as significant predictors of returning to low-risk in 2021 (Appendix A). Of the 61 participants who returned to low-risk, 35 returned to a level of CES-D-10 ≤ 7. Reapplying the complete model to differentiate this subset from the 58 persistent high-risk participants with CES-D-10 ≥ 10 revealed that bedtime remained a significant predictor, but the age-by-coping interaction reduced to a trend level factor (*p* = 0.06, Appendix A).

## 4. Discussion

These novel longitudinal data show divergent trajectories in risk of depression across the COVID-19 pandemic in young people in the U.S two years into the pandemic. Specifically, 38% of participants were resilient (low-risk across surveys), while the remaining young adults split into three high-risk trajectories: continuous high-risk before and during the pandemic (14%), early onset of high-risk from the beginning (June 2020) of the pandemic (30%), and delayed onset of high-risk (18%). These onset rates were beyond the expected increase with aging before the pandemic (average annual onset rate 1.2%). Also novel were the identified factors, i.e., earlier bedtime and better coping skills (for individuals in the mid to late twenties) at the beginning of the pandemic, predicting a return to low-risk of depression one year later.

We found that earlier bedtimes at the start of the pandemic were predictive of a faster decline in depression scores (Figure 2a), which comports with existing studies showing that later bedtime in young adults is associated with depressed mood [41,42]. As also indicated by our analysis, later bedtime was unlikely a pre-existing preference but instead emerged in the context of the pandemic in some individuals. Linked to depression during the pandemic [43], sleep health is a multidimensional construct of which timing is one component [44] and sleep duration is another [45]. Other work has shown a bidirectional relationship between healthy sleep and the reduction and possible prevention of depression [46]. In our prior NCANDA analysis, we showed that pre-pandemic shorter sleep duration predicted a greater increase in CES-D-10 scores at the start of the pandemic [13]. Even though we did not find any evidence that sleep duration predicted a return to low-risk of depression, our findings extend knowledge about the sleep–depression relationship in the context of the pandemic, highlighting the role of integrating good sleep hygiene habits into post-pandemic psychological interventions.

For the older individuals in the sample (mid to late twenties), adaptive coping skills were associated with returning to low-risk of depression, confirming the influence of effective coping strategies for staving off depression [47,48] and recovery of psychopathology [23,24] during crises [29,49]. Notably, coping strategies did not predict a reduction in depression risk in the younger participants, which could reflect developmental differences in coping strategies or may indicate the likelihood that older participants had more experiences in dealing with difficult situations [50]. In addition to the coping strategies measured herein, studies have shown that the association between planning, emotional support seeking, and instrumental support seeking and depressive symptoms changes in the early twenties, with older participants less likely to use denial or avoidance than younger ones [24].

Despite the greater vulnerability of women to the distress of the pandemic than men [8,13,29,51] (as also shown in our prior analysis of the NCANDA sample [13]), our model did not reveal a sex effect in the declining risk trajectory. This finding comports with the results of Wang et al. [29], who reported that, during the pandemic, girls were more likely than boys to have depressed mood; however, there was no sex difference in the trajectory of depressive symptoms in persistent versus risk-declining groups. In addition to sex, our model did not reveal any effect of alcohol drinking in the trajectory of depressive symptoms despite the findings of our prior NCANDA analysis [13] that identified frequent alcohol drinking before the pandemic as predicting risk of depression at the onset of the pandemic.

An open question is the clinical implication of sustained depressed mood for the 28% of NCANDA participants with persistent high-risk from 2020 to 2021. Although no prior study reported data on long lasting high-risk of depression, studies on clinical depression show that the duration of the depression can be detrimental to later brain and psychological functioning of young adults [52], with poorer social functioning for those who suffered from persistent depression, even after recovery [53]. In adults, more persistent major depressive disorder is associated with a lower likelihood of remission, suggesting that more intensive treatment might be necessary for those who seek help after a longer course of elevated symptoms [54]. Of note, many college students experienced higher levels of depressive symptoms and greater disruption to their academic progress and transition to work during the COVID-19 pandemic than did young adults who completed their studies prior to the pandemic [55].

The current study has many strengths, including a longitudinal design capturing within-person changes in depressive symptoms before and across two years of the pandemic and assessment of a range of behavioral and lifestyle factors that are relevant in the context of the pandemic. Nonetheless, the following study limitations should also be considered: Despite using a demographically diverse cohort, the generalizability of the results is limited due to the relatively small sample size and not controlling for COVID-19 exposure across the five NCANDA collection sites or for current local or governmental restrictions (e.g., stay-at-home orders) at those sites. In addition, the self-reported questionnaires with low objectivity might have introduced bias from social desirability and recall period [56] and could not assess the presence of a major depressive disorder, which will require follow-up. Furthermore, other than the sleep variables investigated herein, the evening chronotype is another well-documented risk factor for depression [57,58] and the morning chronotype seems to be a protective factor during the pandemic [59], both of which were not considered in the current analysis. Lastly, while we identified factors predicting recovery of individuals with early onset of depressive symptoms, what remains to be identified is the mechanism for the delayed onset of depressive symptoms in 18% of the participants one year after the outbreak and the factors predicting their recovery.

## 5. Conclusions

In summary, half of the high-risk adolescents reported an improvement in their mental wellbeing in the second year of the pandemic. Alarmingly, however, 28% (compared to 14% before the pandemic) were persistently at high-risk for depression for more than one year, with only a slight improvement in 2021. Our finding that earlier bedtime routines and more adaptive coping (for those in their mid-to-late twenties) buffer the mental health burden of the COVID-19 pandemic could be leveraged to help the most vulnerable young people recover from depression symptoms. These health-promoting behaviors may be useful beyond the COVID-19 pandemic and apply to other public health emergencies or periods marked by intense stress or difficulty [60].

## Figures and Tables

**Figure 1 ijerph-19-10300-f001:**
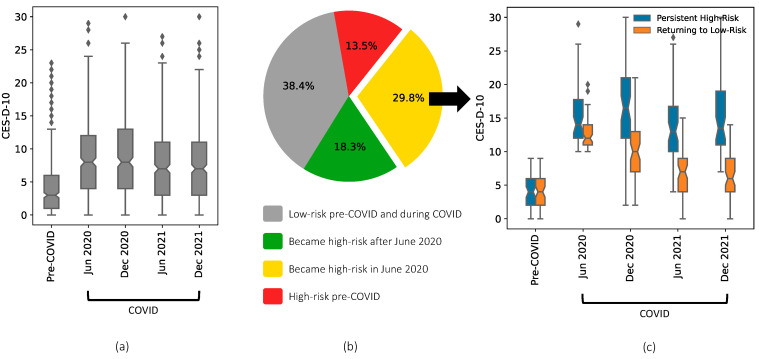
(**a**) The trajectory of depressive symptoms from pre-COVID-19 to December 2021 of 399 NCANDA participants (age 18.0 to 28.6 years in June 2020). The CES-D-10 score increased significantly in 2020 compared to pre-COVID-19 visits (*p* < 0.001) and then significantly declined in 2021 (*p* < 0.001). (**b**) Participants grouped by the time of becoming high-risk (CES-D-10 ≥ 10). (**c**) A total of 51% of the participants who became high-risk in June 2020 returned to the low-risk level in 2021, while the remaining participants stayed high-risk.

**Figure 2 ijerph-19-10300-f002:**
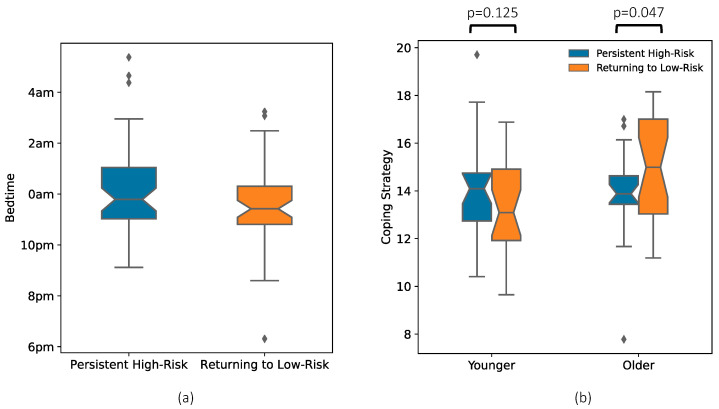
(**a**) Bedtime in June 2020 of the 61 participants returning to the low-risk level was significantly earlier than that of the 58 participants who remained high-risk. (**b**) Older participants returning to low-risk had significantly better coping strategies than the older persistent high-risk group. The same was not true for the younger groups (than the median age at 22.2 years).

**Table 1 ijerph-19-10300-t001:** Demographics of the 399 participants from the NCANDA cohort included in the analysis. Variables were measured at COVID-19 Survey 1 (June 2020).

*Variable*	Mean ± SD or N (Percentage)
Age (years) at COVID-19 Survey 1 (June 2020)	22.8 ± 2.6
Sex	
*Female*	227 (57%)
*Male*	172 (43%)
Family Socioeconomical Status (SES) ^a^	17.0 ± 2.4
Race/Ethnicity	
*Asian*	32 (8%)
*African American/Black*	34 (8.5%)
*Caucasian/White*	296 (74.2%)
*Other*	37 (9.3%)
Hispanic	42 (10.5%)
Study site	
*UC San Diego*	111 (27.8%)
*SRI International*	85 (21.3%)
*Duke University Medical Center*	58 (14.5%)
*University of Pittsburgh*	78 (19.6%)
*Oregon Health & Sciences University (OHSU)*	67 (16.8%)
Alcohol Consumption	
*Number of drinking days in the past 30 days*	5.25 ± 6.57
*Number of drinks in a 24-h period*	1.58 ± 1.72
Pittsburgh Sleep Quality Index items	
*Sleep duration (hours)*	8.82 ± 1.44
*Bedtime (hours)*	11:57 p.m. ± 1.89
*Subjective Sleep quality (0—Very Good, 3—Very bad)*	1.06 ± 0.67
Coping score (Brief Resilient Coping Scale, out of 20)	15.1 ± 2.28

Note: ^a^ highest number of years of education of either parent; SD = standard deviation.

**Table 2 ijerph-19-10300-t002:** Results of the base logistic regression model applied to the 119 participants who exceeded the depression threshold in June 2020.

	*Estimate*	*SE*	*tStat*	*p-Value*
Intercept	0.025	0.579	0.043	0.966
Age	−0.199	0.27	−0.739	0.46
Sex	−0.014	0.202	−0.069	0.945
Age:sex	−0.263	0.226	−1.165	0.244
Study site				
*Duke*	0.256	0.777	0.33	0.741
*OHSU*	0.46	0.618	0.744	0.457
*SRI*	0.801	0.615	1.302	0.193
*UPMC*	0.451	0.679	0.663	0.507
Race				
*Asian*	−0.716	0.717	−0.998	0.318
*African American*	0.204	0.89	0.229	0.819
*Others*	−0.661	0.824	−0.803	0.422
Hispanic	−0.621	0.799	−0.778	0.437
Family socioeconomic status	0.075	0.228	0.328	0.743
Enrolled in education	−0.558	0.495	−1.128	0.26

**Table 3 ijerph-19-10300-t003:** Results of the complete logistic regression model applied to the 119 participants who exceeded the depression threshold in June 2020. Significant predictors are highlighted in bold (*p* < 0.05).

	*Estimate*	*SE*	*tStat*	*p-Value*
Intercept	−3.670	1.390	−2.641	0.008
Age	−0.388	0.412	−0.941	0.346
Sex	−0.242	0.336	−0.720	0.471
Age:sex	−0.801	0.418	−1.917	0.055
Study site				
*Duke*	0.589	1.065	0.553	0.580
*OHSU*	0.032	0.856	0.038	0.970
*SRI*	0.712	0.813	0.876	0.381
*UPMC*	1.187	1.013	1.171	0.242
Race				
*Asian*	0.235	0.953	0.247	0.805
*African American*	−0.857	1.302	−0.658	0.510
*Others*	0.118	1.133	0.104	0.917
Hispanic	**−3.165**	**1.274**	**−2.485**	**0.013**
Family socioeconomic status	−0.213	0.320	−0.666	0.505
Enrolled in education	−1.379	0.753	−1.830	0.067
**CES-D score in June 2020**	**0.316**	**0.091**	**3.464**	**<0.001**
Sleep behavior				
*sleep quality*	−0.131	0.331	−0.395	0.693
** *sleep bedtime* **	**0.919**	**0.387**	**2.373**	**0.018**
*sleep duration*	0.547	0.363	1.507	0.132
*age:sleep quality*	−0.030	0.325	−0.094	0.925
*sex:sleep quality*	−0.304	0.285	−1.067	0.286
*age:sleep bedtime*	0.306	0.329	0.930	0.353
*sex:sleep bedtime*	−0.192	0.345	−0.558	0.578
*age:sleep duration*	−0.344	0.335	−1.028	0.304
*sex:sleep duration*	−0.256	0.317	−0.809	0.418
Coping skills				
*coping*	−0.032	0.303	−0.106	0.916
** *age:coping* **	**−1.224**	**0.476**	**−2.569**	**0.010**
*sex:coping*	0.087	0.308	0.284	0.776
Alcohol use				
*alcohol quantity*	0.276	0.499	0.553	0.580
*alcohol frequency*	0.254	0.518	0.491	0.624
*age:alcohol quantity*	0.626	0.566	1.107	0.268
*sex:alcohol quantity*	0.448	0.427	1.050	0.294
*age:alcohol frequency*	0.130	0.554	0.235	0.814
*sex:alcohol frequency*	0.000	0.472	0.000	0.999

## Data Availability

The analysis is based on the data release NCANDA_PUBLIC_7Y_COVID_REDCAP_V03 [63] made accessible to the public according to the NCANDA Data Distribution [64].

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
