# Peer review of "Earlier Bedtime and Effective Coping Skills Predict a Return to Low-Risk of Depression in Young Adults during the COVID-19 Pandemic"

_ijerph, 2022, doi:10.3390/ijerph191610300_

Round 1

Reviewer 1 Report

In this article, the authors sought to investigate the factors that promoted the reduction in depressive symptoms in high-risk individuals during the COVID-19 pandemic. The authors affirm that earlier bedtime and effective coping skills predict a return to low risk of depression in young adults.

Identifying the factors that could reduce depressive symptoms in high-risk young adults is welcome, and further studies are warranted, given the clinical implications involved in this field. However, minor comments should be addressed before the manuscript is eligible for publication.

Please find my specific comments below and answer point-by-point.

Minor remarks:

Abstract, line 24. Specify the mean and standard deviation of the sample

Introduction: The authors affirm that "Among many other studies documenting an increase in depressive symptoms among young adults during the COVID- 19 pandemic, our prior analysis on the participants from the National Consortium on Alcohol and Neurodevelopment in Adolescence (NCANDA) indicated that risk for depression in young adults tripled in June 2020 compared with the risk for depression in 8 pre-pandemic years. […] Sleep difficulties, insufficient sleep, and eveningness – both pre-pandemic and ongoing during the pandemic– are well-documented risk factors for depressed mood, supporting sleep as a promising modifiable factor for preventing mood deficits and improving depression outcomes".

1 A citation seems necessary at the beginning of the first sentence.

2 During the pandemic emergency, young adults, emerged as a vulnerable category. Indeed, higher rates of sleep problems and mental disorders were reported in this population. Throughout the "Introduction" section, the authors reported a general overview of the major risk factors for depression, presenting only pre-pandemic scientific evidence. However, in the literature, several cross-sectional and retrospective studies have shown that the lockdown period had more pervasive repercussions on sleep and the mental health of young adults, also compared to other high-risk population categories.

In the "Introduction" section, the authors should report the literature about sleep quality/habits and coping strategies and their role in developing depressive symptoms during the pandemic.

Participants and procedure. The pre-COVID period included in the analyses is unclear. Please, specify. Moreover, it would be more intuitive to indicate the pre-covid measurement as Survey 1 and the measurement in June 2020 as Survey 2.

Participants and procedures. Please check the format of all tables in the entire manuscript (i.e., Hispanic).

Participants and procedures. In the capture of Table 1, please correct "age range" with "mean and standard deviation". Moreover, the authors should clarify the "sleep quality" component in the tables. In this manuscript, the authors used the PSQI score to assess sleep behavior. However, I suppose that the authors had used for the analysis the subcomponent of the PSQI "subjective sleep quality". The authors should clarify the component used in all analyses and explain the purpose of this choice.

Discussion. The results obtained by the authors could be ascribable to the biological tendency to morningness chronotype to exhibit an earlier bedtime and, consequently, low vulnerability to depression, reported both in pre-pandemic and pandemic studies (i.e., https://doi.org/10.1038/s41598-021-90993-y) The authors should discuss this aspect in the "Discussion" section. Moreover, they should present the absence of chronotype measures as a study limitation.

Author Response

In this article, the authors sought to investigate the factors that promoted the reduction in depressive symptoms in high-risk individuals during the COVID-19 pandemic. The authors affirm that earlier bedtime and effective coping skills predict a return to low risk of depression in young adults.

Identifying the factors that could reduce depressive symptoms in high-risk young adults is welcome, and further studies are warranted, given the clinical implications involved in this field. However, minor comments should be addressed before the manuscript is eligible for publication.

Please find my specific comments below and answer point-by-point.

Response: We thank the reviewer for the positive feedback and have responded to each concern below.

Minor remarks:

Abstract, line 24. Specify the mean and standard deviation of the sample

Response: We added “age range: 22.8  2.6 years” to the abstract.

Introduction: The authors affirm that "Among many other studies documenting an increase in depressive symptoms among young adults during the COVID- 19 pandemic, our prior analysis on the participants from the National Consortium on Alcohol and Neurodevelopment in Adolescence (NCANDA) indicated that risk for depression in young adults tripled in June 2020 compared with the risk for depression in 8 pre-pandemic years. […] Sleep difficulties, insufficient sleep, and eveningness – both pre-pandemic and ongoing during the pandemic– are well-documented risk factors for depressed mood, supporting sleep as a promising modifiable factor for preventing mood deficits and improving depression outcomes".

1 A citation seems necessary at the beginning of the first sentence.

Response: We have now added two references that showed an increase in depressive symptoms among young adults during the pandemic:

Varma, P., et al., Younger people are more vulnerable to stress, anxiety and depression during COVID-19 pandemic: A global cross-sectional survey. Prog Neuropsychopharmacol Biol Psychiatry, 2021. 109: p. 110236.

Stanton, R., et al., Depression, Anxiety and Stress during COVID-19: Associations with Changes in Physical Activity, Sleep, Tobacco and Alcohol Use in Australian Adults. Int J Environ Res Public Health, 2020. 17(11).

“Among many other studies documenting an increase in depressive symptoms among young adults during the COVID- 19 pandemic (11,12),…”

2 During the pandemic emergency, young adults, emerged as a vulnerable category. Indeed, higher rates of sleep problems and mental disorders were reported in this population. Throughout the "Introduction" section, the authors reported a general overview of the major risk factors for depression, presenting only pre-pandemic scientific evidence. However, in the literature, several cross-sectional and retrospective studies have shown that the lockdown period had more pervasive repercussions on sleep and the mental health of young adults, also compared to other high-risk population categories.

In the "Introduction" section, the authors should report the literature about sleep quality/habits and coping strategies and their role in developing depressive symptoms during the pandemic.

Response: As suggested, we modified this paragraph to highlight existing works on sleep and coping strategies in relation to depressive symptoms during the pandemic and highlighted differences between younger and older individuals.

“Poor sleep quality [19] and disrupted sleep, longer sleep latency, and higher daytime dysfunction [20] were prevalent among young adults during the COVID-19 outbreak contributing to their higher levels of depression than reported in older adults. Sleep difficulties are well-documented risk factors for depressed mood, implicating sleep as a promising modifiable factor for preventing mood deficits [21] and improving depression outcomes [22]. Another crucial factor in the prevention and recovery of psychopathology in young people that could be critical in the context of the pandemic is adaptive coping strategies in response to life stress [23-25]. Longitudinal analyses of 1,251 adolescents [24] revealed that the association between poorer coping strategies and depressive symptoms increased after the age of 17.5 years. In a sample of 881 first-year college students in Netherland, maladaptive coping (such as denial, venting, substance use, and self-blame) mediated the trajectory of depression and anxiety at the beginning of the pandemic [26]. Compared to older adults, these different coping strategies of younger individuals had negative effects, rendering them vulnerable to developing psychopathology and stressful events [27].”

The new references are:

Hyun, S., et al., Psychological correlates of poor sleep quality among U.S. young adults during the COVID-19 pandemic. Sleep Med, 2021. 78: p. 51-56.

Amicucci, G., et al., The Differential Impact of COVID-19 Lockdown on Sleep Quality, Insomnia, Depression, Stress, and Anxiety among Late Adolescents and Elderly in Italy. Brain Sci, 2021. 11(10).

Marshall, A., et al., Resilience to COVID-19: Socioeconomic Disadvantage Associated With Higher Positive Parent-youth Communication and Youth Disease-prevention Behavior. Res Sq, 2021 DOI: 10.21203/rs.3.rs-444161/v1.

Freyhofer, S., et al., Depression and Anxiety in Times of COVID-19: How Coping Strategies and Loneliness Relate to Mental Health Outcomes and Academic Performance. Front Psychol, 2021. 12: p. 682684.

Fukase, Y., et al., Age-related differences in depressive symptoms and coping strategies during the COVID-19 pandemic in Japan: A longitudinal study. J Psychosom Res, 2022. 155: p. 110737.

Participants and procedure. The pre-COVID period included in the analyses is unclear. Please, specify. Moreover, it would be more intuitive to indicate the pre-covid measurement as Survey 1 and the measurement in June 2020 as Survey 2.

Response: We added Figure S1 in the supplement plotting the distribution of the Pre-COVID visit date and further clarified the definition of the pre-COVID visits in the following sentences.

“For each participant, the most recent visit before March 1, 2020 was considered their pre-COVID assessment… ”

“… The pre-COVID visits of these 399 adults ranged from July 2013 to February 2020 with the median date in June 2019.”

In line with the naming convention in our prior work [13], we prefer to identify this visit as “Pre-COVID”, so that it is immediately clear in the Figures and analyses what is the impact of COVID. It also minimizes confusion with the multiple surveys conducted during COVID times.

Participants and procedures. Please check the format of all tables in the entire manuscript (i.e., Hispanic).

Response: We reformatted Table 1 and note that the Hispanic variable is a separate variable from the race label according to the NCANDA data protocol “Brown, S.A., et al., The National Consortium on Alcohol and NeuroDevelopment in Adolescence (NCANDA): A Multisite Study of Adolescent Development and Substance Use. J Stud Alcohol Drugs, 2015. 76(6): p. 895-908”.

Participants and procedures. In the capture of Table 1, please correct "age range" with "mean and standard deviation". Moreover, the authors should clarify the "sleep quality" component in the tables. 

Response: We added the age range and clarified the sleep quality component in Table 1

In this manuscript, the authors used the PSQI score to assess sleep behavior. However, I suppose that the authors had used for the analysis the subcomponent of the PSQI "subjective sleep quality". The authors should clarify the component used in all analyses and explain the purpose of this choice.

Response: Thank you for this recommendation. We have now clarified that the NCANDA COVID-19 surveys used only a subset of items from the PSQI (Pittsburgh Sleep Quality Index) (Buysse et al., 1989), such as the item about sleep quality.

“Participants also completed the subset of items from the PSQI (Pittsburgh Sleep Quality Index) [38] that asked about bedtimes, wake-up times, sleep duration and sleep quality in the past month. Sleep quality was assessed with the question, “During the past month, how would you rate your sleep quality overall?”, with options ranging from very good to very bad. The choice of using this single rating to encode sleep quality is due to its strong correlation with the total PSQI score in the NCANDA sample (Hasler et al., 2017).”

Discussion. The results obtained by the authors could be ascribable to the biological tendency to morningness chronotype to exhibit an earlier bedtime and, consequently, low vulnerability to depression, reported both in pre-pandemic and pandemic studies (i.e., https://doi.org/10.1038/s41598-021-90993-y) The authors should discuss this aspect in the "Discussion" section. Moreover, they should present the absence of chronotype measures as a study limitation.

Response: Indeed, our results are in line with the literature reporting that eveningness is a risk factor for mood disorders both prior (Merikanto et al., 2013) and during the pandemic (Salfi et al., 2021). Thank you for the suggestion, which we have added to our discussion, and we also note that the lack of a chronotype measure is a limitation of the analysis.

“Also, other than the sleep variables investigated herein, the evening-chronotype is another well-documented risk factor for depression [46, 47], while the morning chronotype seems to be a protective factor during the pandemic [48], both of which were not considered in the current analysis.”

Reviewer 2 Report

Congratulations to the authors for the work and effort. 

I proposed some minimal changes in the format of the article. 

It would be convenient to explain Table 1 better.

This table has two columns, where one column reads 18.0 and the other 28.6. It is not very easy to understand what the authors want to reflect with this table.

I would advise authors not to highlight parts of the article text in bold or underline, as this is not usually done in scientific articles.

It would be interesting in figure 1, that the authors analyzed High risk pre-covid and the group Became high risk after June 2020, in the same way that they analyze the other two groups.

Author Response

Congratulations to the authors for the work and effort. I proposed some minimal changes in the format of the article. 

Response: We thank the reviewer for the positive feedback and have responded to each concern below.

It would be convenient to explain Table 1 better.

This table has two columns, where one column reads 18.0 and the other 28.6. It is not very easy to understand what the authors want to reflect with this table.

Response: We now reformatted Table 1 to improve clarity and readability.

I would advise authors not to highlight parts of the article text in bold or underline, as this is not usually done in scientific articles.

Response: We removed bold and underline in the manuscript.

It would be interesting in figure 1, that the authors analyzed High risk pre-covid and the group Became high risk after June 2020, in the same way that they analyze the other two groups.

Response: We kindly note that the analysis was specifically targeted on those who became high risk at the beginning of the pandemic (yellow pie in Figure 1) to identify factors (in June 2020) predicting their recovery in 2021. The persistent high-risk group (red pie) already had high depressive symptoms before COVID so the analysis on them would be irrelevant to the pandemic. The late-onset group (green pie) was still at high-risk in 2021 so analyzing their recovery status would require the 2022 data which is still under collection. Nevertheless, we added the following to the discussion.

“Lastly, while we identified factors predicting recovery of individuals with early onset of depressive symptoms, what remains to be identified is the mechanism for the delayed onset of 18% of the participants one year after the outbreak and the factors predicting their recovery.”